# Evaluation of Optimized Toluidine Blue Stain as an Alternative Stain for Rapid On-Site Evaluation (ROSE)

**DOI:** 10.3390/diagnostics15101223

**Published:** 2025-05-13

**Authors:** Alejandro S. Mendoza, Alaa Afify, Lydia Howell, John Bishop, Aurelia Lauderdale, Stan Seko, Ronelson Hermosilla, Donald York, Kurt B. Schaberg

**Affiliations:** Department of Pathology and Laboratory Medicine, UC Davis Health, Sacramento, CA 95817, USAkbschaberg@ucdavis.edu (K.B.S.)

**Keywords:** telecytology, FNA, ROSE, TEA stain, Toluidine Blue (TB), Diff-Quik (DQ)

## Abstract

**Background**: Rapid on-site evaluation (ROSE) is crucial for improving the diagnostic yield of fine-needle aspiration (FNA) biopsies. Despite recent advances in ROSE, such as telecytology, the rapid stains used in this process have not seen significant innovation. Diff-Quik (DQ) and Toluidine Blue (TB), the most common ROSE stains, have significant limitations. This study evaluates the optimized Toluidine Blue stain, a mixture of **T**oluidine Blue, **E**osin, and **A**lcohol (TEA), as a potential alternative to TB or DQ for ROSE. **Methods**: A comparative study was conducted using fifty remnant body fluid specimens with adequate cellularity, collected at the University of California Davis Medical Center over six months. Two smears were prepared from each specimen. One was stained with TB, and the other with optimized Toluidine Blue (TEA). Digital images of each slide were evaluated by three cytologists and two cytopathologists, blinded to the stain, using five criteria: background staining, cytoplasmic detail, nuclear membrane clarity, chromatin texture, and nucleoli staining. Each criterion was scored on a scale of 1 to 3. **Results**: Optimized Toluidine Blue (TEA) stain demonstrated superior overall image quality compared to TB. Specifically, optimized Toluidine Blue (TEA) showed significantly less background staining (*p* < 0.05) and improved nuclear membrane clarity (*p* < 0.05), chromatin texture (*p* < 0.05), and nucleoli detail (*p* < 0.05). There was no significant difference between the two stains in the assessment of cellularity or cytoplasmic detail. **Conclusions**: The optimized Toluidine Blue (TEA) stain shows promise as a rapid stain for ROSE, offering rapid processing and improved digital image quality. Further evaluation of optimized Toluidine Blue (TEA) stain on FNA specimens is warranted to validate these findings and explore its potential to enhance telecytology in ROSE.

## 1. Introduction

Rapid on-site evaluation (ROSE) is a vital procedure performed during fine-needle aspiration (FNA) biopsy. FNA biopsy is a minimally invasive technique used to obtain tissue samples for diagnostic examination. ROSE involves the immediate, real-time microscopic assessment of these FNA samples, performed while the patient is still undergoing the procedure. This immediate evaluation allows the clinician to determine if an adequate sample has been obtained, ensuring that sufficient material is available for an accurate diagnosis. By reducing the rate of inadequate samples, ROSE significantly improves the diagnostic yield and minimizes the need for repeat biopsies. FNA biopsies without ROSE have reported inadequacy rates of 20–25%, often leading to repeat procedures that increase costs for patients, hospitals, and insurers, and delay patient care [1,2,3,4,5,6].

Commonly used stains in ROSE, including Diff-Quik (DQ), Toluidine Blue (TB), and rapid Pap stain [7], have individual advantages and disadvantages. The choice between these stains often depends on the personal preference of the practitioner, shaped by their training and experience. At UC Davis Health, TB has been favored for over 30 years due to its efficiency, economy, and practicality [8,9,10,11,12]. TB reduces specimen preparation time and simplifies stain management. However, unlike the multi-chromatic DQ stain, TB provides a relatively monochromatic appearance, which can make interpretation challenging and necessitates significant user experience. TB may also inadequately penetrate areas with thick mucus and blood, limiting its effectiveness in some specimens.

In addition to traditional microscopy, telecytology has improved the accessibility of cytopathology expertise for ROSE. Telecytology refers to the practice of evaluating cytological specimens remotely. This is achieved through the use of digital imaging technologies that capture images of the specimen, which are then transmitted to a cytopathologist at a different location for interpretation [13,14,15,16,17,18,19,20,21]. Telecytology is particularly valuable in situations where on-site cytopathology expertise is not readily available, such as in rural or remote areas, or when multiple procedures occur simultaneously. While telecytology has expanded access to specialized diagnostic services, innovation in the rapid stains used in ROSE has been limited.

This study introduces a new staining process, the optimized Toluidine Blue stain, designed to enhance the cellular detail observed with TB while maintaining its efficiency in terms of processing time and material requirements. The optimized Toluidine Blue stain utilizes a staining kit composed of **T**oluidine Blue, **E**osin, and **A**lcohol (TEA). This study presents an evaluation of TEA stain, comparing its interpretability to that of TB in the digital evaluation of cytology specimens. We hypothesize that optimized Toluidine Blue (TEA) stain will provide comparable or superior image quality to TB for the rapid assessment of cellularity and nuclear detail, crucial for ROSE adequacy assessment.

## 2. Materials and Methods

### 2.1. Study Design and Sample Selection

This comparative study evaluated the performance of a new rapid stain, optimized Toluidine Blue (TEA), against the conventional Toluidine Blue (TB) stain in the assessment of cytologic material. Fifty deidentified remnant body fluid specimens were selected from the cytology laboratory storage at UC Davis Health over a 6-month period (July to December 2019). Body fluids were chosen for this initial study due to their availability, allowing for the preparation of additional smears without affecting patient care. To ensure adequate material for evaluation, specimens were included if they had been screened for sufficient cellularity and had been stored for less than two weeks. This storage time limit was implemented to minimize potential degradation of cellular morphology. An aliquot of each specimen was centrifuged, and the concentrate was examined to confirm adequate cellularity prior to inclusion in this study.

### 2.2. Specimen Preparation and Staining

For each of the 50 selected specimens, two smears were prepared. Both smears were fixed in alcohol. One smear was then stained with TB, and the other with optimized Toluidine Blue. The optimized Toluidine Blue stain comprises two solutions, containing a mixture of eosin with alcohol (Solution #1) and Toluidine Blue (Solution #2). The optimized Toluidine Blue staining procedure involves the following steps: (1) application of Solution #1 to the slide (by spraying or dipping) for 3 seconds, (2) application of Solution #2 (by drops or dipping) for 3 seconds, and (3) coverslipping. The slide is then ready for immediate microscopic evaluation. There is no time difference to stain using either TB or optimized TB, and both can be completed in less than 10 seconds from smearing to completion of stain application.

### 2.3. Image Acquisition

Representative digital images were captured from the stained smears using an Olympus BX51 microscope (Olympus Corporation, Tokyo, Japan) equipped with a digital camera. To standardize image acquisition, the microscope’s illumination settings were pre-set, and Cellsens software (Ver 3.1.1) was used with pre-set automatic exposure settings. Images were taken from multiple fields of view to ensure that the captured images were representative of the overall slide.

### 2.4. Image Evaluation

The digital images were evaluated by five reviewers: two cytopathologists and three cytologists. The reviewers were blinded to the identity of the stain (TB or TEA) for each image. The reviewers first assessed the adequacy of the material in the images for diagnostic interpretation. They then evaluated the image quality of both TB- and optimized Toluidine Blue (TEA)-stained images using the following five criteria, which are critical for ROSE assessment:Background staining: the amount of residual stain in the background.Cytoplasmic detail: the clarity and distinctness of cytoplasmic features.Nuclear membrane clarity: the sharpness and definition of the nuclear membrane.Chromatin texture: the distinctness and pattern of chromatin within the nucleus.Staining of nucleoli: the visibility and prominence of nucleoli.

Each criterion was scored on a scale of 1 to 3, where 1 represented the poorest quality and 3 the highest quality (Table 1).

### 2.5. Statistical Analysis

Interobserver agreement for the assessment of cellularity was calculated using Fleiss’ Kappa statistic. Differences in image quality scores between TB and optimized Toluidine Blue (TEA) stains for each of the five criteria were assessed using the paired Student’s *t*-test. All statistical analyses were performed using Microsoft Excel (Redmond, WA, USA). A *p*-value of less than 0.05 was considered statistically significant.

## 3. Results

There was substantial inter-rater agreement on the assessment of cellular adequacy for both TB- and optimized Toluidine Blue (TEA)-stained specimens (Fleiss’ Kappa = 0.78). While optimized Toluidine Blue (TEA)-stained specimens were rated slightly higher for cellularity than TB-stained specimens, this difference was not statistically significant (*p* = 0.34). Overall agreement between raters was 85% (Table 2).

The optimized Toluidine Blue (TEA) stain demonstrated a cleaner background compared to TB (Figure 1 and Figure 2). Specifically, optimized Toluidine Blue (TEA)-stained images showed significantly less residual background staining than TB-stained images (*p* < 0.05). There was no significant difference between the two stains in the quality of cytoplasmic detail (*p* = 0.85) (Figure 1 and Figure 3).

However, optimized Toluidine Blue (TEA) staining resulted in significantly improved nuclear membrane clarity (*p* = 0.03) (Figure 1 and Figure 4), chromatin texture (*p* = 0.005) (Figure 1 and Figure 5), and nucleoli staining (*p* = 0.0002) (Figure 1 and Figure 4) compared to TB staining. Nuclear membranes were sharper and better defined with optimized Toluidine Blue (TEA), and chromatin texture was easier to evaluate. Nucleoli were also more prominent and distinct in optimized Toluidine Blue (TEA)-stained images.

## 4. Discussion

The current study reinforces the well-established importance of ROSE in FNA procedures. Previous studies have consistently demonstrated that ROSE improves diagnostic yield and reduces inadequacy rates [1,2,3,4,5,6]. This study further supports the continued need for effective ROSE procedures and techniques.

The findings of the current study, which demonstrate improved nuclear detail and reduced background staining with the optimized Toluidine Blue (TEA) stain compared to TB, contribute new insights to the literature on rapid staining techniques. While previous studies have compared different stains for ROSE [7], this study provides a detailed evaluation of a novel optimized Toluidine Blue stain and its potential advantages.

This study’s use of Toluidine Blue (TB) as a comparator is consistent with previous studies that have evaluated TB in various cytological applications. Some studies have reported on the effectiveness of TB for rapid staining [8], acknowledging its limitations, such as monochromatic staining, which can make interpretation challenging. The current study builds upon this existing knowledge by presenting an optimized version of TB to address these limitations.

The optimized Toluidine Blue (TEA) stain itself is a novel contribution. This study provides a detailed evaluation of this new staining method, offering a potential alternative to existing stains. This study’s systematic evaluation of specific image quality criteria (background staining, cytoplasmic detail, nuclear membrane clarity, chromatin texture, and nucleoli staining) provides a more detailed and objective assessment of stain performance compared to some previous studies. This study explicitly addresses the implications of the findings for telecytology, highlighting the importance of optimized staining for improved remote diagnosis.

The discussion of telecytology in the context of ROSE reflects the increasing recognition of telecytology’s crucial role in modern cytopathology. Telecytology, the practice of remote cytological evaluation, has emerged as a transformative tool, extending the reach of expert diagnostic services beyond the physical confines of traditional laboratories [16,17,18,19,20,21]. This is particularly important in scenarios where immediate on-site evaluation by a cytopathologist is not feasible, such as in rural hospitals, during off-hours, or when multiple urgent cases occur simultaneously. As telecytology continues to evolve with advancements in digital imaging and communication technologies, its role in ROSE and cytopathology is poised to expand further, driving innovation in diagnostic practices and improving patient outcomes.

This study’s emphasis on the need for high-quality digital images for telecytology aligns with previous work that consistently highlights the paramount importance of image quality for accurate remote diagnosis. In telecytology, the cytopathologist or cytologist relies solely on the digital representation of the specimen, making image clarity, resolution, and color fidelity essential for identifying subtle cellular details and making accurate interpretations. Poor image quality can lead to misdiagnosis, delayed reporting, and increased anxiety for both clinicians and patients.

This study highlights the potential of the optimized Toluidine Blue (TEA) stain to address limitations in current ROSE staining methods, particularly within the context of digital cytology and the expanding field of telecytology. By providing improved image quality in digital representations, the TEA stain offers a promising tool for enhancing diagnostic confidence and efficiency in both on-site and remote ROSE procedures, ultimately contributing to improved patient care.

## 5. Conclusions

This study demonstrates that the optimized Toluidine Blue (TEA) stain is a promising alternative rapid stain for ROSE, particularly in the context of telecytology. Optimized Toluidine Blue (TEA) staining provides image quality that is comparable or superior to that of TB, especially in terms of background staining, nuclear membrane clarity, chromatin texture, and nucleoli detail.

Future research should evaluate the performance of optimized Toluidine Blue (TEA) stain on FNA specimens to better simulate typical ROSE conditions and to assess the impact of these background elements on staining quality. Additional studies should also explore the long-term stability of TEA-stained slides and compare its performance to other rapid staining methods, including DQ and rapid Papanicolaou stains. Finally, additional studies are needed to determine whether the improved image quality with optimized Toluidine Blue (TEA) stain translates to improved diagnostic accuracy and efficiency in ROSE procedures.

## Figures and Tables

**Figure 1 diagnostics-15-01223-f001:**
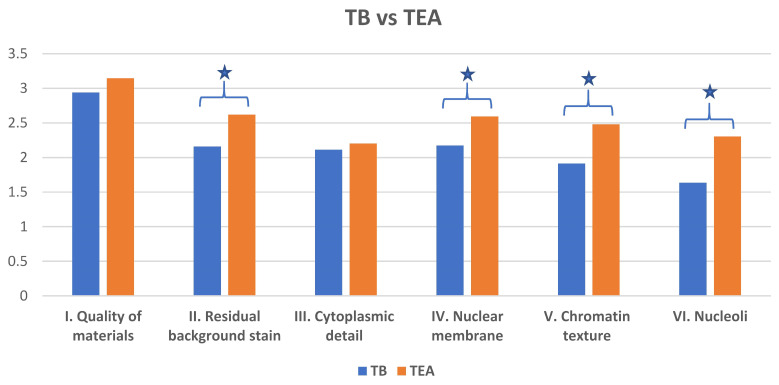
Mean reported image quality for TB vs. TEA staining for all six evaluated metrics. ★ *p* < 0.05 (paired *t*-test).

**Figure 2 diagnostics-15-01223-f002:**
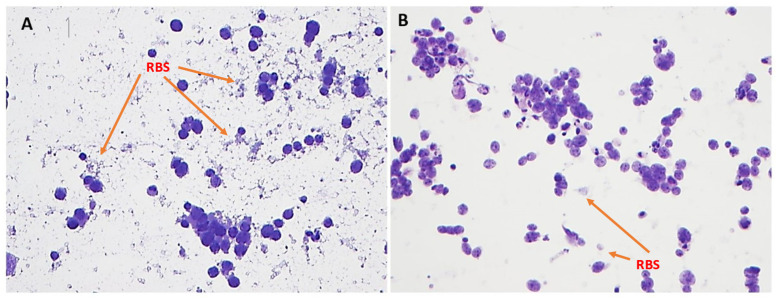
Residual background stain (RBS). TB (**A**, left) versus TEA (**B**, right). Significantly less background staining was seen with the TEA stain (100× magnification).

**Figure 3 diagnostics-15-01223-f003:**
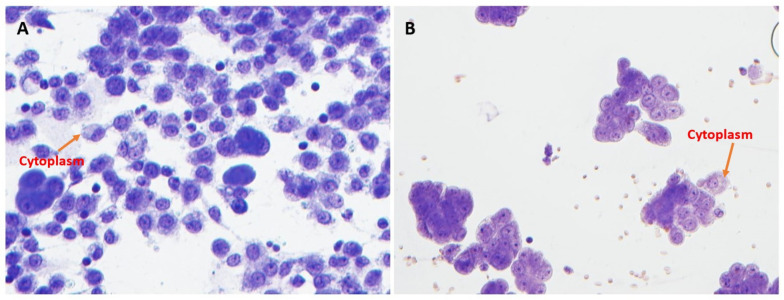
Cytoplasmic details of TB (**A**, left) versus TEA (**B**, right). No significant differences were seen with cytoplasmic detail (400× magnification).

**Figure 4 diagnostics-15-01223-f004:**
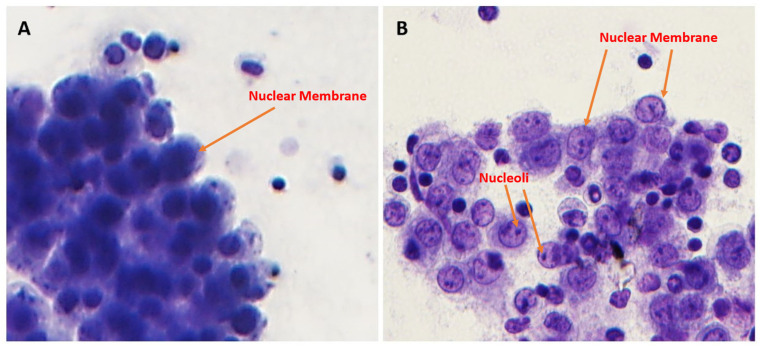
Nucleoli and nuclear membrane details. TB (**A**, left) versus TEA (**B**, right). Nucleoli and nuclear membrane visibility were significantly better with the TEA stain (600× magnification).

**Figure 5 diagnostics-15-01223-f005:**
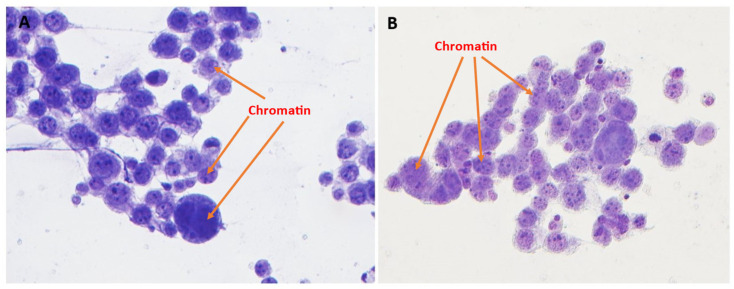
Chromatin texture. TB (**A**, left) versus TEA (**B**, right). Chromatin texture was significantly easier to evaluate with the TEA stain (400× magnification).

**Table 1 diagnostics-15-01223-t001:** Categories and criteria used for grading stain quality. A score of 1 to 3 could be assigned to each category. For each, the lowest score is 1 and the highest score is 3.

	QUALITY SCORES		
CATEGORY	1	2	3
Nuclear membrane	not distinct	some details	distinct
2. Chromatin texture	not distinct	some details	distinct
3. Nucleoli	not distinct/absent	some details	distinct
4. Cytoplasmic detail	not distinct	some details	distinct
5. Residual background stain	dirty	mild-moderate presence	absent/clean

**Table 2 diagnostics-15-01223-t002:** Inter-raters’ concordance on the quality of material examined.

Fleiss’ Kappa (>2 rater reliability/concordance)	
Percent overall agreement	85%
Free-marginal kappa	0.70
95% CI for free-marginal kappa	0.52, 0.88

## Data Availability

Limited data are available upon request.

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
