# Peer review of "Evaluation of Optimized Toluidine Blue Stain as an Alternative Stain for Rapid On-Site Evaluation (ROSE)"

_diagnostics, 2025, doi:10.3390/diagnostics15101223_

Round 1

Reviewer 1 Report

Comments and Suggestions for Authors The manuscript titled 'Evaluation of Optimized Toluidine Blue Stain as an Alternative Rapid Stain for Rapid On-Site Evaluation (ROSE)' compares traditional Toluidine Blue stain with optimized Toluidine Blue stain in body fluids to improve digital image quality. The results favor Optimized Toluidine Blue; however, certain aspects require further detail.   Introduction: Please define ROSE (Rapid On-Site Evaluation) and telecytology to provide context for readers. Materials and Methods: 1. Given that the samples were collected and stored between July - December 2019, what is the rationale for publishing this study in 2025, considering the significant time gap? 2. Specify whether the optimized Toluidine Blue stain was prepared in the laboratory or is commercially available. 2 (a). If commercially available, provide the catalogue number, company, and country name according to journal guidelines. 2 (b). If prepared in the laboratory, describe the protocol for preparing Solution 1 and Solution 2, including concentrations of alcohol, eosin, and Toluidine Blue. Additionally, clarify whether these solutions are prepared fresh each time or can be stored for weeks/months, and specify any storage conditions. 3. Clarify whether the body fluids stained for TB and optimized TB were the same or different. 4. Please mention the time required to perform both optimized Toluidine Blue and traditional Toluidine Blue staining procedures to highlight any potential time-saving benefits. 5. Please provide justification of including 5 (3 cytologists and 2 cytopathologists) when diagnostic accuracy was not achieved in FNAC samples. Results: 1. Replace Figures 5 and 6 with clearer images that include chromatin and nucleoli. 2. Add annotations to all figures for clarity. 3. Please include inter observers' findings/data as supplementary file. Discussion: Provide a more comprehensive discussion by comparing the current study's results with previous studies, highlighting similarities and differences. References: Please standardize the referencing style format by either including or excluding month consistently across all references, as per the journal's guidelines.  

Author Response

Comments 1:  Introduction: Please define ROSE (Rapid On-Site Evaluation) and telecytology to provide context for readers.

Responses 1:See revised Introduction (pages 1-2):

Rapid On-Site Evaluation (ROSE) is a vital procedure performed during fine needle aspiration (FNA) biopsy.  FNA biopsy is a minimally invasive technique used to obtain tissue samples for diagnostic examination. ROSE involves the immediate, real-time microscopic assessment of these FNA samples, performed while the patient is still undergoing the procedure.  This immediate evaluation allows the clinician to determine if an adequate sample has been obtained, ensuring that sufficient material is available for accurate diagnosis.  By reducing the rate of inadequate samples, ROSE significantly improves diagnostic yield and minimizes the need for repeat biopsies.  FNA biopsies without ROSE have reported inadequacy rates of 20-25%, often leading to repeat procedures that increase costs for patients, hospitals, and insurers, and delay patient care (1–6).   

Commonly used stains in ROSE, including Diff-Quik (DQ), Toluidine blue (TB) and rapid Pap stain (7), have individual advantages and disadvantages.  The choice between these stains often depends on the personal preference of the practitioner, shaped by their training and experience.  At UC Davis Health, TB has been favored for over 30 years due to its efficiency, economy, and practicality (8–12).  TB reduces specimen preparation time and simplifies stain management.  However, unlike the multi-chromatic DQ stain, TB provides a relatively monochromatic appearance, which can make interpretation challenging and necessitates significant user experience.  TB may also inadequately penetrate areas with thick mucus and blood, limiting its effectiveness in some specimens.   

In addition to traditional microscopy, telecytology has improved the accessibility of cytopathology expertise for ROSE.  Telecytology refers to the practice of evaluating cytological specimens remotely.  This is achieved through the use of digital imaging technologies that capture images of the specimen, which are then transmitted to a cytopathologist at a different location for interpretation (13-21).  Telecytology is particularly valuable in situations where on-site cytopathology expertise is not readily available, such as in rural or remote areas, or when multiple procedures are occurring simultaneously.  While telecytology has expanded access to specialized diagnostic services, innovation in the rapid stains used in ROSE has been limited.   

This study introduces a new staining process, the optimized Toluidine Blue stain, designed to enhance the cellular detail observed with TB while maintaining its efficiency in terms of processing time and material requirements.  The optimized Toluidine Blue stain utilizes a staining kit composed of Toluidine blue, Eosin, and Alcohol (TEA).  This study presents an evaluation of TEA stain, comparing its interpretability to that of TB in the digital evaluation of cytology specimens.  We hypothesize that optimized Toluidine Blue (TEA) stain will provide comparable or superior image quality to TB for the rapid assessment of cellularity and nuclear detail, crucial for ROSE adequacy assessment. 

Comments 2:  Materials and Methods:  Given that the samples were collected and stored between July - December 2019, what is the rationale for publishing this study in 2025, considering the significant time gap?

Responses 2:  The primary reason for this extended timeframe is that the corresponding author initiated the study and manuscript preparation but had to pause the work due to a period of employment in private practice.  The corresponding author has since returned to UC Davis and has been able to finalize and submit the manuscript.  While the data is from 2019, the findings remain relevant as they address a persistent need for improved staining techniques in ROSE and telecytology, which continue to be important in current practice.

Comments 3:  Specify whether the optimized Toluidine Blue stain was prepared in the laboratory or is commercially available.

Responses 3: At the time of the study, the stain kits were prepared in the laboratory.

Comments 4: If prepared in the laboratory, describe the protocol for preparing Solution 1 and Solution 2, including concentrations of alcohol, eosin, and Toluidine Blue. Additionally, clarify whether these solutions are prepared fresh each time or can be stored for weeks/months, and specify any storage conditions.

Responses 4:  The specific concentrations of alcohol, eosin, and Toluidine Blue in the optimized Toluidine Blue (TEA) stain are considered proprietary information and constitute a trade secret. Therefore, we cannot disclose the precise formulation.

However, we can provide the following clarification: The optimized Toluidine Blue (TEA) stain is now commercially available, as of May 1, 2025, through a third-party vendor: (Catalogue #TEAK-50, ScyTek Laboratories, USA). https://www.scytek.com/products/48-TEAK-50-TEA-Stain-Kit.asp#

At the time of the study, the prepared kits were stored and had a shelf life of one year.

Comments 5:  Clarify whether the body fluids stained for TB and optimized TB were the same or different.

Responses 5: Each body fluid sample was stained with TB and optimized TB for comparison. 

Comments 6: Please mention the time required to perform both optimized Toluidine Blue and traditional Toluidine Blue staining procedures to highlight any potential time-saving benefits.

Responses 6: There is no time difference to stain using either TB or optimized TB, and both can be completed in less than 10 seconds from smearing to completion of stain application. (Added to 2.2. Specimen Preparation and Staining, page 3).

Comments 7: Please provide justification of including 5 (3 cytologists and 2 cytopathologists) when diagnostic accuracy was not achieved in FNAC samples. 

Responses 7: Cytologists and cytopathologists bring complementary expertise to the evaluation of cytologic material. Cytologists are highly skilled in the initial screening and identification of cellular components, while cytopathologists possess advanced training in diagnostic interpretation and the recognition of subtle pathological changes. Including both groups allowed for a more comprehensive assessment of the staining quality, encompassing both technical and diagnostic perspectives.

In clinical practice, both cytologists and cytopathologists are involved in the ROSE procedure. Cytologists often perform the initial slide preparation and screening, providing a first-line assessment of adequacy, while cytopathologists make the final diagnostic interpretations. Our study design mirrors this real-world workflow, enhancing the translatability of our findings.

While the study did not aim to assess diagnostic accuracy in the context of FNA samples, it focused on evaluating the image quality of the stains. The criteria used for evaluation (background staining, cytoplasmic detail, nuclear membrane clarity, chromatin texture, and nucleoli staining) are fundamental aspects of slide quality that both cytologists and cytopathologists routinely assess. Therefore, their combined input was valuable in providing a well-rounded evaluation of the stains' performance.

Comments 8:  Results: 1. Replace Figures 5 and 6 with clearer images that include chromatin and nucleoli. 2. Add annotations to all figures for clarity. 3. Please include inter observers' findings/data as supplementary file. 

Responses 8: See changes made to the figures (pages 4-5). See observers’ data attached as supplementary file.

Comments 9:  Discussion: Provide a more comprehensive discussion by comparing the current study's results with previous studies, highlighting similarities and differences. 

Responses 9:  See revised discussion (pages 6-7). The current study reinforces the well-established importance of ROSE in FNA procedures. Previous studies have consistently demonstrated that ROSE improves diagnostic yield and reduces inadequacy rates (1-6). This study further supports the continued need for effective ROSE procedures and techniques.

The findings of the current study, which demonstrate improved nuclear detail and reduced background staining with the optimized Toluidine Blue (TEA) stain compared to TB, contribute new insights to the literature on rapid staining techniques. While previous studies have compared different stains for ROSE (7), this study provides a detailed evaluation of a novel optimized Toluidine Blue stain and its potential advantages.

The study's use of Toluidine blue (TB) as a comparator is consistent with previous studies that have evaluated TB in various cytological applications. Some studies have reported on the effectiveness of TB for rapid staining (8), acknowledging its limitations, such as monochromatic staining, which can make interpretation challenging. The current study builds upon this existing knowledge by presenting an optimized version of TB to address these limitations.

The optimized Toluidine Blue (TEA) stain itself is a novel contribution. The study provides a detailed evaluation of this new staining method, offering a potential alternative to existing stains. The study's systematic evaluation of specific image quality criteria (background staining, cytoplasmic detail, nuclear membrane clarity, chromatin texture, and nucleoli staining) provides a more detailed and objective assessment of stain performance compared to some previous studies. The study explicitly addresses the implications of the findings for telecytology, highlighting the importance of optimized staining for improved remote diagnosis.

The discussion of telecytology in the context of ROSE reflects the increasing recognition of telecytology's crucial role in modern cytopathology. Telecytology, the practice of remote cytological evaluation, has emerged as a transformative tool, extending the reach of expert diagnostic services beyond the physical confines of traditional laboratories (16-21). This is particularly important in scenarios where immediate on-site evaluation by a cytopathologist is not feasible, such as in rural hospitals, during off-hours, or when multiple urgent cases occur simultaneously. As telecytology continues to evolve with advancements in digital imaging and communication technologies, its role in ROSE and cytopathology is poised to expand further, driving innovation in diagnostic practices and improving patient outcomes.

The study's emphasis on the need for high-quality digital images for telecytology aligns with previous work that consistently highlights the paramount importance of image quality for accurate remote diagnosis. In telecytology, the cytopathologist or cytologist relies solely on the digital representation of the specimen, making image clarity, resolution, and color fidelity essential for identifying subtle cellular details and making accurate interpretations. Poor image quality can lead to misdiagnosis, delayed reporting, and increased anxiety for both clinicians and patients.  

This study highlights the potential of the optimized Toluidine Blue (TEA) stain to address limitations in current ROSE staining methods, particularly within the context of digital cytology and the expanding field of telecytology. By providing improved image quality in digital representations, the TEA stain offers a promising tool for enhancing diagnostic confidence and efficiency in both on-site and remote ROSE procedures, ulti-mately contributing to improved patient care.

Comments 10:  References: Please standardize the referencing style format by either including or excluding month consistently across all references, as per the journal's guidelines.  

Responses 10:  References updated (pages 8-9). Months removed. 

Reviewer 2 Report

Comments and Suggestions for Authors

Dear Authors, 

 In my opinion, your manuscript entitled Evaluation of Optimized Toluidine Blue Stain as an 
Alternative Rapid Stain for Rapid Onsite Evaluation (ROSE) is interesting. The idea is perfect. The speed of diagnosis is, in most cases, significant for surgeons and patients. The manuscript is not well-written. The discussion is not good. They are only information about results; this is not a discussion. There are no references to literature. Please correct this. The references list is the most interesting articles about cytodiagnostics, stains, and HE. Please discuss with them.

Regards

Author Response

Comments 1: The manuscript is not well-written. The discussion is not good. They are only information about results; this is not a discussion. There are no references to literature. Please correct this. The references list is the most interesting articles about cytodiagnostics, stains, and HE. Please discuss with them.

Responses 1: Agree.  We have revised the manuscript, expanded the discussion and now include a comprehensive comparison of our findings with existing literature on ROSE, rapid staining techniques, and telecytology (pages 6-7). We have specifically incorporated relevant references from the provided list, as well as other pertinent studies, to contextualize our results within the broader field of cytodiagnostics. This expanded discussion aims to highlight the similarities and differences between our findings and previous research, emphasizing the novel contributions of our optimized Toluidine Blue (TEA) stain.

We believe that these revisions have significantly strengthened the discussion, providing a more robust interpretation of our results and their implications for clinical practice and the evolving landscape of telecytology.

Thank you again for your constructive criticism, which has been invaluable in improving our manuscript. We are confident that the revised version now addresses your concerns.

Round 2

Reviewer 1 Report

Comments and Suggestions for Authors

The comments have been addressed and manuscript has been revised.

Reviewer 2 Report

Comments and Suggestions for Authors

Dear Authors, 

 The second version of your manuscript is perfect! Thank you very much for  taking my comments into consideration

Best regards